# Mental health inequalities, challenges and support needs during COVID-19: a qualitative study of 14-to-25-year-olds in London

Jasmine Lee , Keri Ka-Yee Wong

Psychology and Human Development, University College London, London, UK

**Correspondence to**
Dr Keri Ka-Yee Wong;
keri.wong@ucl.ac.uk

## ABSTRACT

**Objectives** The impact of the COVID-19 pandemic on adolescent's mental health and relationships has received growing attention, yet the challenges and support needs of adolescents living in existing deprivation are not well understood. The current qualitative study, part of a broader project cocreating mental health and life-skill workshops with young people, documents adolescents' lived experience and support needs 4 years on from the COVID-19 pandemic.

**Design** 20 semi-structured interviews and 6 focus groups were transcribed and thematically analysed in NVivo V.12 to inform codesigned workshops to support adolescents' needs.

**Setting** Islington borough in North London, United Kingdom.

**Participants** 20 adolescents aged 14–25 years (mean=18.3; 60% female, 60% white) from Islington with a history of difficulties (eg, mental health, deprivation, court order) were referred by Islington local authority teams to our study.

**Results** Thematic analyses revealed eight themes on adolescents' COVID-19 experiences and five associated suggestions on 'support needs': health challenges and support; relationships and support; routines and support; educational challenges and learning support; inequality and support; distrust; loss of opportunities and grief.

**Conclusions** In our qualitative study, adolescents spoke of positive reflections, challenges, and need for support 4 years on from the COVID-19 pandemic. Many adolescents shared their lived experiences for the first time with someone else and wished they would have the space and time to acknowledge this period of loss. Adolescents living with existing inequality and deprivation before the pandemic have reported sustained and exacerbated impacts during the pandemic, hence coproduced support for adolescents should be a priority.

## STRENGTHS AND LIMITATIONS OF THIS STUDY

⇒ Participants felt they had an opportunity to open up about experiences of the pandemic and cocreate workshop content for other young people.
⇒ Baseline sociodemographic deprivation was assumed and not explicitly measured as recruitment was based on Islington Council referrals of young people most at-risk, and not in education, employment or training young people were not reached.
⇒ One-off brief interviews and focus groups may underestimate the impact of COVID-19 on adolescents.

experiences and access to support are no longer discussed, hence lessons learnt are forgotten.

Adolescents aged 10–24 years[4] have been particularly hard-hit by COVID-19—impacting an important developmental period of biological, psychological and social change.[5] Estimates of 135 global studies[6] found higher rates of mental health issues in 4–18 years than previously thought: anxiety (31%; n=1 241 604 vs 21%), depression (31%; n=524 417 vs 25%), and sleep disturbances (42%; n=104 219).[7] Consistent with prepandemic findings, higher levels of depression and anxiety were found in girls (than boys), older (than younger) and earlier 2020–2021 (than 2021–2022) parts of the pandemic.

In particular, COVID-19's longer-term impact on adolescents experiencing inequalities and deprivation[8]—for example, gender, race/ethnicity, socioeconomic status, health inequalities, what support looked like and what is needed now are understudied. Moreno-Agostino *et al*'s secondary data analysis study of the 1990 and 2000–2002 British birth cohorts (N=8588) between February and March 2021 reported large mental health inequalities across sexual minority status, with females and adolescents (teens/20s) reporting the worst outcomes on anxiety, depression, loneliness and life satisfaction.[9]

## INTRODUCTION

COVID-19 has brought into sharp focus society's inequalities.[1] Four years on, COVID-19 continues to impact lives[2] with an ever-widening gap between groups living with/without inequalities and deprivation (eg, pre-existing health conditions and minoritised groups[3]). Conversations about pandemic

Another longitudinal study (March 2020–May 2021) of UK 4–16 years also found 14%–31% experienced high and increasing mental health difficulties, and were more likely to have special education needs/neurodevelopmental disorders, parents with higher psychological distress, parent–child conflict, and poorer peer relationships compared with their peers.[10] Understanding how various inequalities impact adolescent's mental health can inform current and future public health policies, and importantly, practical solutions.

Qualitative studies of adolescents' mental health and support needs, particularly from minority ethnic communities, are extremely limited. A coproduced qualitative systematic review of COVID-19's impact on 10–24 years mental health (N=50 731; UK=74%) found five themes: pandemic information and restrictions; education and learning; social connection; emotional, lifestyle and behavioural changes; and mental health support, yet adolescent perspectives minoritised by ethnicity, sexual orientation and marginalised/vulnerable groups were missing.[11] In a focus group study, Burgess *et al*[12] examined COVID-19's impact on 16–25 years ethnic minority's mental health (black=87%); themes included deteriorating mental health, experiences of racism and examples of resilience/growth through self-care practices. Only one other semistructured interview study of 12–17 years from black and mixed ethnic groups (N=10; females=70%) conducted in a deprived part of West London during October–December 2021 focused on challenges and support needs, finding: (1) worsening mental health, self-doubt, loneliness, disordered eating; (2) coping skills, resilience, new connections and (3) limited COVID-19 support, support for trust, health and skills development.[13] Though a non-representative sample, adolescents' first-hand experiences of inequality and deprivation during the pandemic and unmet support needs highlight a priority area for research and practice.

### Aims and objectives

The current interview and focus group study examines COVID-19's impact on 14–25 year-olds living with inequality and deprivation and the support needs for this group. We engaged with young people living in a highly deprived borough of London and referred by youth workers, in order to better understand the pandemic experiences and support needs of those disproportionately affected by the pandemic due to their socioeconomic backgrounds, disabilities, carer duties and more. By actively including them in interviews and focus groups, codesigning of workshops, as well as knowledge dissemination, we aimed to provide participants with a platform to voice their concerns in a way that they feel greater ownership over the research project.

### METHODS

This preregistered study published on the Open Science Framework in advance (Wong, 2022)[14] is part of the larger Empower-Islington partnership between University College London and Islington Council (https://osf.io/82pgm/).[15] The Empower-Islington project aims to identify, cocreate and assess the efficacy of life-skill workshops with Islington youth to maximise their life chances post-COVID-19. After completing the qualitative component (the current study), five expert-led workshops were codeveloped with young people to address the support needs brought up.

### Patient and public involvement

There was public, young people, involvement throughout the study interviews, focus groups, recruitment and codesign of workshops appropriate to the target sample. Young people and parents were also invited to take part in disseminating study findings at an academic conference, to feedback on their experience at regular Islington Youth Council meetings, and in the production of the end-of-project animation and report with the animator, council and researchers.

### Participants

Islington is among the nation's 20% most deprived neighbourhoods, third most dangerous in London (15%+crime) and has high unemployment rate in 18–24 years (57.3% vs rest of Britain, 39.8%).[16] 52% of residents identify as black Asian minority ethnic. As such, adolescents aged 14–25 years who are Islington residents with a history of difficulties (eg, mental health, socioeconomic deprivation, court order, carer duties) and who may benefit most from additional mental health/life-skill support were identified by council and youth club social workers and referred to us based on the highest need for support (eg, not in education, employment or training). Parental consent was obtained from all under-18s and parents were welcome to stay and wait in a different space during interviews and focus groups to ensure inclusivity for those with additional needs.

20 adolescents aged 14–25 years (mean=18.3, SD=3.05 years, no. of females=12) consented and completed a baseline mental health survey prior to a recorded one-on-one 20–30 min (average=26 min) in-person (n=13) or Zoom (n=7) interview. Of the 20 participants, 17 attended one of seven researcher-led focus groups (average group=2.83; duration=22.5 min), in-person (n=3) or on Zoom (n=3). On completion, participants were debriefed and received a £20 e-voucher (£10 interview, £10 focus group) of their choice. Participants self-identified as being white (n=12), black or black African (n=3), Bangladeshi (n=2), Indian (n=1), white and Asian (n=1), or from other ethnic groups (n=1) (online supplemental appendix A). Our sample was a predominantly white and female sample, suggesting that our workshops may be more appealing to this group and more efforts to recruit a more diverse and gender-balanced sample are needed in the future studies. However, it also does not reflect the young men who were referred but were not captured in our study for other reasons.

## Student researchers

Student researchers (including JL) were recruited via UCL channels, trained to conduct peer research and paid for their time. Seven researchers (second-year undergraduates=5 and master's=2) conducted the interviews and focus groups, supported by the principal investigator KK-YW. Meetings were held throughout the project to ensure researchers received necessary support and to discuss their experiences of the research process.

In light of Braun and Clarke's emphasis on reflexivity in thematic analysis,[17] we reflected on the power imbalance that exists between our team of researchers and youth participants who often faced multiple inequalities, acknowledging that our own privileges may have influenced data collection, analysis and interpretation. For example, our team of researchers, although somewhat diverse in terms of ethnicity, gender and sexual orientation, had not faced the same socioeconomic deprivation as our participants, and this privileged lens may have precluded us from depicting the true quality of their experience. Participants may also have felt less comfortable to share authentically.

## One-on-one interviews

A 15-item semi-structured interview (online supplemental appendix B) first designed by KK-YW and further refined by student researchers was administered to assess adolescents' lived experiences and support received (or lack thereof) (6 questions) during/after the pandemic. Prods helped guide interviewers.

## Focus groups

Building on challenges identified in one-on-one interviews, the focus groups averaging 22.5 min asked: 'What workshop would you be interested in participating in?' 'What support would you like (eg, health, routines and habits, education, and employment)?'.

## Analysis

Researchers transcribed the interviews and focus groups verbatim and thematically analysed the data in NVivo V.12, using Braun and Clarke's six-step framework to identify common themes and support needs.[18] JL (primary coder) read and re-read the transcripts, developed a set of codes until no new codes were added further, and reached theoretical saturation after 10 transcripts. Final codes were discussed and confirmed (KK-YW) (online supplemental appendix C). Three new coders were trained and conducted line-by-line coding on a new transcript to establish excellent inter-rater reliability ($\kappa > 0.86$). All coders independently coded the remaining transcripts followed by final checks.

## RESULTS

From the interviews and focus groups, we clustered the 20 adolescents' COVID-19 experiences into 8 themes, of which support needs were identified for five (figure 1 and online supplemental appendix D). Not central to current study aims are other non-COVID-19 developmental difficulties that are not discussed (online supplemental appendix E).

### Theme 1: health

The pandemic impacted adolescents' mental health (n=19) and physical health (n=17). While eight participants spoke positively about introspection and growth during COVID-19 ('*to reflect on myself and mature*', 14–17 years old, male), self-doubt and deterioration in self-image and body-image, largely driven by social media use and comparison with others, plagued over half of the participants' experiences (n=11).

*I used to be quite confident… now it's like I had to re-learn all of that… after the pandemic* (18-21 years old, gender undisclosed)

*the whole like 'lockdown body challenge'… it affected a lot of things for me, like in terms of body image…* (22-25 years old, female)

Around one-third of participants (n=7) struggled with loneliness ('*trapped and lonely*', 18–21 years old, female) because of lockdown restrictions. 'Lack of access to nature', not walking around parks was challenging for seven participants, with only two talking about having access to green spaces. Not being able to '*move outside of the four walls*' of home fuelled feelings of confinement, and not '*going out and exercising*' (18–21 years old, gender undisclosed) impacted participant's physical health (n=17) – for example, '*I put on weight during Covid… now I am out of breath more*' (14–17 years old, female), sleep patterns, and binge eating (n=7) (see theme 3).

### Health support

Adolescents wanted support on mental and physical health challenges. Participants called for improved mental health provisions for adolescents (n=18) and school access to mental health services and resources (n=12) (eg, 'mental health workshops' and 'learning mentors' support), recognising the current long waiting lists in services.

*everything was such a backlog and long waiting list. It just wasn't something I could access.* (18-21 years old, female)

Participants wanted a compulsory meeting with a '*coach or adviser*' for '*mental health and life advice*' (22–25 years old, gender undisclosed), especially for individuals at-risk for mental illnesses like depression. One-on-one and '*group therapy*' may also provide relief to participants as they know others are '*feeling like how they feel*' (14–17 years old, male).

*everyone should be offered mental health and life advice at least once… 'This is your mentor, you can take up this service or not'* (22-25 years old, gender undisclosed)

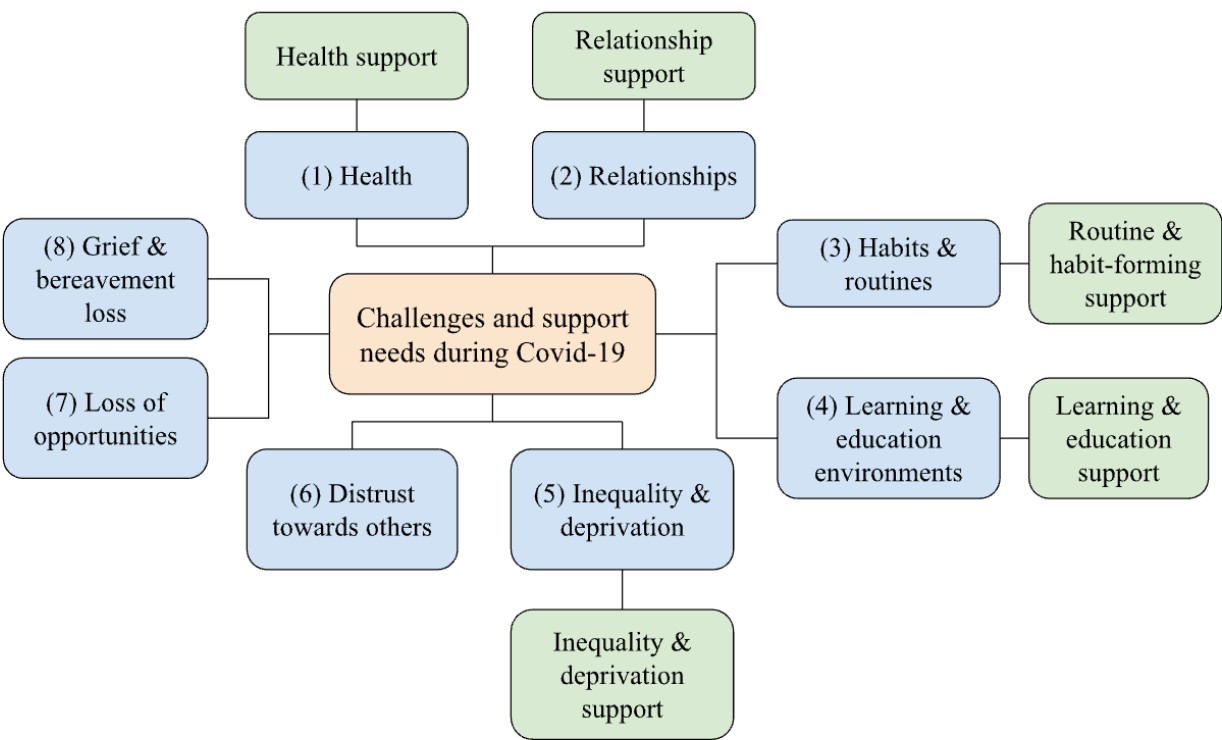

**Figure 1** A concept map showing eight emerging themes (blue), five of which include support needs (green). Support needs for three themes (6–8) were not brought up in discussions with young people and hence not included.

Around one-third (n=7) of participants were eager to be heard. Participants found speaking to researchers about COVID-19 to be valuable: '*I haven't really spoken to anyone [about Covid], except like, you… this is the best time I've spoken of Covid*' (14–17 years old, female). While the general public seemed eager to '*move on*' and '*forget… that COVID-19 was even a thing*' (18–21 years old, gender undisclosed), participants desired to talk and process experiences with someone, and to hear that others went through the same, '*providing a space to talk about this experience because we pretend like it's now in the past, even though it's still something actually very present*' (22–25 years old, female).

More specific support on building confidence and self-esteem (n=7) was mentioned. Participants proposed workshops to build social skills and learn to avoid comparison with others on social media ('*confidence… building up the social skills and the confidence to go to [places like gyms]*', 14–17 years old, male). For some, simply having someone encourage them and say, '*you can do it*' (14–17 years old, female) would have been helpful.

Linking mental and physical health, one-third (n=7) of participants wanted support for physical health, especially on improving sleep, exercise and eating habits, '*more exercise rather than sitting around*' (14–17 years old, female) or having access to '*a gym for young people*' (14–17 years old, male) to support a more active lifestyle, even during restrictions.

## Theme 2: relationships

18 of 20 participants spoke about friendship changes during COVID-19, diluting friendships and fall-outs due to '*lack of contact and… communication*' (14–17 years old, female), as well as forming new positive and healthier relationships after a period of reflection and change.

> *the friends that I had at the time… we couldn't hang out in person. So we just then ended up not really texting and just drifted apart.* (14-17 years old, female)
>
> *I think covid was very useful in showing me who my true friends were and who weren't* (14-17 years old, female)

Sustaining friendships became more difficult. Relying on technology to maintain social contact was difficult for participants facing digital exclusion, '*I didn't have a phone… I never got those numbers.*' (18–21 years old, female) The 'tone of voice' online also caused misunderstandings, '*texting… led to me gaining more friends but also me losing a lot of friends because… not being able to understand tone in conversations*' (18–21 years old, male).

Unsurprisingly, COVID-19 also impacted family life (n=16). Some adolescents were stuck at home with family much more than they were used to and conflict ensued, '*during Covid being cooped up in my house, with just my family was really tiring for me and it made us prone to more arguments…*' (14–17 years old, male) while other participants (n=9) became closer to their family, '*I could spend*

*more time with family, I liked that'* (22–25 years old, gender undisclosed).

## Relationship support

Friends were good sources of support for many (n=14). They communicated mostly via text and calls, but participants would have wanted more friends to motivate and support them through schoolwork and feelings of loneliness during lockdown.

*friends were really a big help when it came to stuff like learning, the loneliness so like psychological impact…* (14-17 years old, female)

Family support was important, particularly on schoolwork during online learning and conversations around emotional regulation when locked down indoors—this was especially crucial for those living away from family.

*I definitely needed support from my parents, who, despite being ill, were able to help me significantly. It really gave me that passion and drove me towards finishing all my homework.* (14-17 years old, female)

Although many participants appreciated family support (n=8), it is likely that many refrained from sharing experiences of complicated home conditions, perhaps due to lack of family support. A few shared with researchers only after the recorded interview session, when more rapport was built.

## Theme 3: habits and routines

Many adolescents (n=18) struggled with forming healthy habits disrupted by lockdown, and many still find this challenging to reverse at the time of the interviews.

*it's got me into quite a few habits that involve being sedentary… a lot of lying in bed during Covid.* (18-21 years old, male)

*My biggest problem now is being normal, having a routine, getting up early* (22-25 years old, undisclosed)

Adolescents recognised common bad habits including insomnia, overeating, not exercising and unhealthy habits used to cope with lockdown and escape reality.

*I just couldn't get back into the routine of being up in the day, sleeping at night, and I still can't sleep, I struggle sleeping…* (14-17 years old, female)

*I had a bit of a drug problem… the vulnerability of being stuck indoors, we didn't have anything else to do.* (22-25 years old, gender undisclosed)

## Routine and habit-forming support

Adolescents wanted support on forming healthy habits, hobbies and gaining new skills conducive to healthy lifestyles. They wanted help *'getting into routine'* and *'set[ting] a set structure'* (22–25 years old, male). Two participants specifically wanted help with *'not getting addicted to things'* (22–25 years old, gender undisclosed), like social media. One participant suggested a workshop to truly understand

the impact of social media on health, *'comprehensive understanding of… what it is'* (22–25 years old, male) and how to prevent addiction. Still others spoke about developing new hobbies and healthy routines, *'I spent my free time on a device or just like been eating junk food… I could've been making something… some kind of music, some kind of art form, that would've been really helpful for me.'* (14–17 years old, male)

## Theme 4: challenges with learning and education environments

School and university goers found online learning and education environments challenging (n=14). They struggled with concentration and *'a big lack of motivation'* (14–17 years old, female), resulting in a sense of failure to learn during COVID-19.

*a challenge was… definitely the work I was given. We was given much more work than usual, and using a computer to really do all of that…* (14-17 years old, female)

*I just felt a bit like trapped at uni, like I-I knew what I wanted to do, and I was like depressed and didn't want to be there.* (22-25 years old, female)

Participants were also dissatisfied with the education environments and the lack of teacher support. One participant dropped out of university altogether after being locked in university dorms with a lack of support and resources available.

## Learning and education support

Online learning challenges during COVID-19 highlighted learning gaps that forced participants to catch up quickly. Participant focus groups called for support on improving study skills, productivity, learning and career planning, particularly for those who are *'confused on what they wanna do'* (18–21 years old, male). Students felt ill equipped during the pandemic.

*I think they should have supported students and done a lot better, we didn't have access to the libraries, equipment, communal areas or anything like that.* (22-25 years old, gender undisclosed)

## Theme 5: distrust towards others

Participants were distrustful towards the government, institutions and other people since COVID-19, citing dissatisfaction over the UK government's COVID-19 response and policies, *'the whole Boris Johnson thing… that was very hypocritical… he can't miss a Christmas party, but I had to miss my friend'* and *'you can't trust anyone within government… I don't trust anyone.'* (18–21 years old, female). Broader social and relational mistrust was also raised by half of the participants, due to fake news and misinformation. Participants were confused by whom to trust and what to believe. For some, daily interactions involved being 'very cautious', 'standoffish' and 'cold' (18–21 years old, female).

*Deep down people don't trust anyone. I think because of online, a lot of hidden truths have been exposed.* (22-25 years old, gender undisclosed)

Participants did not explicitly discuss support needs for this theme.

### Theme 6: inequality and deprivation

Although participants were actively recruited through existing experiences of inequalities and deprivation, only eight explicitly discussed inequalities experienced by them or those around them during the pandemic (eg, technological access, carer duties). It is likely that participants faced more difficulties than what they felt comfortable sharing with researchers. For example, digital poverty and exclusion for those without smartphone and laptop access (n=2), and how these impact learning and relationships. Three participants also discussed caring responsibilities for family during the pandemic (eg, caring for a depressed grandmother, a pregnant mum, a father ill with COVID-19) with two participants casually mentioning caring duties of siblings, eg, by default of being the eldest in a family of eight children outside the recorded interview.

*the people that don't have technology that wanted to go on the team would get laptops from school… I think in Islington we did it really well.* (18-21 years old, female)

*I had my nan next door… make sure she was alright at night, obviously with her depression and stuff… there was a lot of us in the house… the house is quite small as well.* (14-17 years old, female)

Family financial struggles were very real to young people, fuelling feelings of insecurity and a need for independence.

*both of my parents lost their jobs due to COVID… that was difficult for them… That was a big hit because I have to defend for myself now.* (22-25 years old, gender undisclosed)

Others spoke of dire learning environments, '*a dingy school*' (18–21 years old, male), and living conditions, for example, a '*very old house*' with '*rattl[ing] windows*' that '*almost fell out*' when sirens sounded (14–17 years old, female).

#### Inequalities and deprivation support

Participants wished the government would '*support more vulnerable people in the community*' (18–21 years old, female). Our participants faced various inequalities—socioeconomic background, disability, youth offending history—and called for more holistic financial, educational and psychological support for adolescents: '*mental and financial support… the pandemic has triggered a cost-of-living crisis… students get nothing… I think they should financially support people*' (22–25 years old, gender undisclosed). For example, '*schools giving out Chromebooks, so that… students can learn at home*' (14–17 years old, female).

### Theme 7: loss of opportunities

Of those interviewed, six participants said COVID-19 resulted in a loss of social and career opportunities.

Participants felt they missed out on key developmental milestones, including social events and once-in-a-lifetime trips abroad. Two participants in particular were disappointed about the loss of job opportunities, which forced them to work even harder afterwards to secure more opportunities.

*I didn't really get to socialise much… it kind of impacted the way I made friends.* (14-17 years old, male)

*I came to do a thing, and I couldn't do the thing that I wanted to do…* (22-25 years old, male)

The pandemic disrupted normal university routines (eg, to '*go clubbing*' with friends, 22–25 years old, gender undisclosed), which turned into an opportunity to self-reflect on their actions instead of blindly following social norms. Participants did not explicitly discuss support needs for this theme.

### Theme 8: grief and bereavement loss

Half of the participants discussed difficulties coping with grief and wanted support. Due to lockdown, adolescents who lost loved ones lost valuable time together before their passing, '*I lost two of my other grandparents this year in very short space… Covid definitely affected my last years with them,*' (18–21 years old, male) and '*after losing someone beloved to them… being able to speak out about it… about how their mental health could've have been ruined due to the pandemic is… really useful*' (14–17 years old, female). Constant news on deaths was also distressing, '*Millions lost their lives due to Covid and it was such an upsetting time*' (14–17 years old, female). Participants did not mention sources of support specific to coping with grief.

### DISCUSSION

The findings of this qualitative research corroborates earlier studies of the COVID-19 pandemic.[11–13] Participants raised similar pandemic themes to previous studies of ethnic minorities and marginalised 10–24 years[11]: mental health support; emotional, lifestyle and behavioural changes; social connection; education and learning and pandemic information and restrictions. Unsurprisingly, adolescents broadly experienced challenges in similar domains of life, but the magnitude of these challenges coupled with existing inequalities varied across adolescents' experiences and support needs.[13]

Health challenges were common. Participants wanted access to mental health services via schools and access to gyms to support physical health during the pandemic to minimise the health deterioration. Some wanted a trusted 'mentor' or 'life coach' to give advice, support and reassurance—a theme that resonates with the impact of COVID-19 fake news linked with poorer mental health outcomes.[19] Distrust in leaders and others impacted relationships and behaviours, in line with research on sustained distrust on poor mental health via loneliness,[20] particularly for adolescents with existing adversities.[21] Distrust in political leadership among young people

may also have led to non-compliance of social distancing guidelines and other pandemic responses.[22] This distrust has substantial implications on health messaging and government policies, and warrants further research, specifically when designing interventions with marginalised communities.

Participants also spoke about addressing inequalities of the digital divide, struggling to trust and relate to others in society's transition to 'normality', and developing practical skills to gain motivation and a sense of routine (both physically and mentally) in learning and working environments to maximise future life outcomes. Financing learning resources for families most in need, facilitating relational support networks that rebuild adolescent's trust in others, alongside low-intensity school-based/community-based interventions, may address adolescents' needs.

A unique and rather unexpected finding was adolescents' experience of participation. Participants felt heard and understood when 'talking about their COVID-19 experiences', many for the first time, and brainstorming ways in which support needs could be met. Participants shared with the team after interviews/focus groups in informal conversations that they had enjoyed and felt comfortable sharing with the research team. Perhaps participants felt more comfortable sharing with peer researchers—who were of similar age—and were more willing to share genuine experiences.

While no group comparisons can be made due to small samples across participant characteristics, this qualitative study captures the lived experience of a sample of predominantly young women with existing inequalities during COVID-19 (eg, financial struggles, care duties) who wish to be heard but are seldom represented in research. Future research and intervention focusing on groups experiencing specific inequalities (eg, those in contact with youth justice service, with carer duties) may lead to greater impact and gains in young people's wellbeing.

Our study highlights the importance of including young people's voices in research, particularly those who are less represented in the literature and marginalised in society, but also of adolescent-centred pandemic prevention work,[23] particularly for individuals living with inequalities and disabilities.[24]

### Strengths and limitations

The study has several strengths. First, participants expressed ownership in the research project and shared support ideas enthusiastically—they engaged not only in interviews/focus groups but also collaborated with the team to create five subsequent mental health and life-skill workshops, as well as with the dissemination of findings. Second, one-on-one interviews in-person and online gave participants flexibility to participate without the social pressure of crossing county lines—areas of the borough where criminal activities (eg, drug crime) are known to take place. They also created an opportunity for young people to open up about their personal experiences, which fuelled a sense of shared experience in finding COVID-19 experiences challenging (ie, during focus groups).

There are also limitations. First, one-off brief interviews and focus groups may underestimate the magnitude of COVID-19's impact on adolescents. Second, baseline deprivation was assumed and not explicitly measured as we relied on Islington Council referrals of at-risk adolescents—those most at-risk were most unresponsive. Third, although participants were involved in interviews, focus groups, workshop building and knowledge dissemination process, they were not involved in the processes of funding, project design and qualitative analysis. In the future, we hope to collaborate with young people in every stage of a project to better achieve the aims of coproduction. Fourth, the wide age range 14–25 years reflected unique developmental challenges, recognising a need to map out age-appropriate support. While we note that the similarity in ages between student research assistants and the current sample of young people may have helped build rapport, future studies leveraging existing relationships already built between caseworkers and young people at regular meetings, and involving young people throughout the early design phases of the project may maximise participation in workshops from those most in need and often neglected.

### Potential implications

This research informs mental health research and practice, existing educational support and ways of working with adolescents on pandemic recovery policies. Key COVID-19 challenges identified by adolescents for adolescents highlight priority areas for support in coproduced research—to involve adolescents through every stage of a research project to reflect their needs; in mental health provisions in schools; evidence-based practices in community services and addressing the digital divide at the national level—all inequalities raised early on in the pandemic[25] but are still not yet addressed or addressed more long term.[26] Other unresolved yet important challenges identified by adolescents include support with regaining trust in others and institutions, loss of career and social opportunities, grief and trauma. Places of work and learning should acknowledge these less observable concerns and provide provisions for support and staff training - all areas of policy considerations.

### CONCLUSION

COVID-19's impact on adolescents' lives and support needs, especially those experiencing sustained inequalities and deprivation, deserves more attention. Coproduced solutions and adolescent-focused pandemic planning across settings (eg, school, help-seeking, policy) may avoid further widening of the inequality-deprivation gap.

**Acknowledgements** The authors would like to thank the Youth Employment Team at Islington Council for actively referring adolescents to take part in this study (Siobhan Scantlebury and Roopa Doshi), the Empower-Islington research team (Isil Bastug, Eoin Mulholland, Jaimie Leung, Martha Oakes, Yao Yuan, Edouard De Bray) for conducting all interviews and focus-groups with such professionalism and last but not least, all participants and families for sharing their experiences and ideas with us on this important topic.

**Contributors** KK-YW was involved in the project conception, securing funding, planning, supervision of data collection, analysis and interpretation, writing and reviewing of manuscript. JL was involved in data collection (interviews and focus groups), transcription, data analyses, writing and reviewing the manuscript. JL and KK-YW agree to be joint guarantors of the work conducted and accountable for all aspects of the work in ensuring that questions related to the accuracy or integrity of any part of the work are appropriately investigated and resolved.

**Funding** This work was supported by UK Research and Innovation, from UCL's HEIF Knowledge Exchange and Innovation Fund 2022–2023 to KK-YW.

**Competing interests** None declared.

**Patient and public involvement** Patients and/or the public were involved in the design, or conduct, or reporting, or dissemination plans of this research. Refer to the Methods section for further details.

**Patient consent for publication** Not applicable.

**Ethics approval** This study involves human participants and was approved by UCL Institute of Education Ethics Committee (REC 1735), data protection number Z6364106/2022/11/128. Participants gave informed consent to participate in the study before taking part.

**Provenance and peer review** Not commissioned; externally peer reviewed.

**Data availability statement** Data are available in a public, open access repository. OSF. https://osf.io/82pgm/.

**ORCID iDs**
Jasmine Lee http://orcid.org/0000-0002-3925-5778
Keri Ka-Yee Wong http://orcid.org/0000-0002-2962-8438

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
