## [Reviewer comments · BMJ Open]

This paper was submitted to a another journal from BMJ but declined for publication following peer review. The authors addressed the reviewers' comments and submitted the revised paper to BMJ Open. The paper was subsequently accepted for publication at BMJ Open.

ARTICLE DETAILS

TITLE (PROVISIONAL)	The mental health inequalities, challenges, and support needs during Covid-19: a qualitative study of 14-25-year-olds in London
AUTHORS	Lee, Jasmine; Wong, Keri Ka-Yee

VERSION 1 – REVIEW

REVIEWER	Cheetham, Mandy Northumbria University, Nursing, Midwifery and Health
REVIEW RETURNED	25-Oct-2023

GENERAL COMMENTS	Thank you for the opportunity to review this paper, which has an important potential contribution to make to the literature given the focus on young people with experience of inequalities. The findings highlight important learning about the health-related, social, educational, employment and familial effects of Covid-19 on young people. However, in my view, major revisions are required before it is ready for publication. I attach detailed comments and feedback below, which I hope are helpful in revising the paper. Stated Aims The current interview and focus group study examines Covid-19's impact on 14-25-year-olds living with inequality and deprivation and co-produced support needs for this group. This is an important focus but I am unclear how inequality and deprivation are defined in the context of the study? Can you say more about the criteria for selection? Or what you mean by co-produced support needs? Can you define co-production and explain in what sense the study was co-produced with young people? It's unclear why and how participants felt ownership of the research project, as is claimed, when they were 'informed of the study aims' rather than developing these jointly with the research team. Methods Further detail would be helpful on how the study fits with the larger Empower-Islington partnership between University College London and Islington Council (https://osf.io/82pgm/) which aims to identify, co-create, and assess the efficacy of life-skill workshops with Islington's 16-25-year-olds to maximise their life chances post-COVID. How and why were young people selected / recruited? What are the implications of the sample characteristics? e.g 12 women. Can you explain what is meant by a pre-registered study? Design
---

	It would be useful to include a clear rationale for the choice / order of methods used. What were the strengths and limitations of having undergraduate and masters students conduct interviews and focus groups? Were they trained as peer researchers? What support did they receive before, during and after data collection? How were they (student researchers) recruited? Ethics of care? Some bold claims are made about the ability to build rapport on the basis of age similarity. What about other experiences / inequalities, including gender, race/ethnicity, socioeconomic status, ability, sexual orientation? What is meant by 'sharing genuine experiences'? Ethics Did young people under 18 provide assent? How did you manage the ethics of working with young people experiencing inequalities? Public involvement Aside from participating in interviews and focus groups, I am unclear how young people were involved, beyond helping with dissemination activities. If young people were not involved in the funding, design, RQs, analysis or interpretation of findings, please say why. If they were involved, please say how. Data analysis What codes were generated? Could you include the coding framework in the appendix? I recommend Braun and Clark (2022) more recent book which encourages reflexive thematic analysis. I'd be interested in hearing more about the researchers' values, politics and positionality and how this influenced data collection and analysis and writing up. Themes I was unclear how the themes were generated. For example, why support needs were not identified for all the themes, including grief / bereavement loss and loss of opportunities? The findings have important implications for policy and practice. How were young peoples' experiences differentiated by age, gender, ethnicity, experience of or contact with the CJS? Could these nuances be drawn out more explicitly in the findings and reflected in the discussion? I would have welcomed further discussion about the interesting and important findings about trust / distrust of politicians and what this means for public health messaging and pandemic responses in the future. Findings As a general rule, in small qualitative studies, it makes more sense to refer to numbers of participants rather than 90% or 60%. (as you do on P 13) P14 refers to 'suboptimal welfare' without an explanation of what this means. P17 lines 29-31. I don't understand the reference to county lines here. Can you explain? Discussion The discussion seems very short and doesn't appear to position the contribution the paper makes in existing literature or fully explore recommendations for practice. As noted, some reflections about the strengths and limitations of (peer research) methods would be helpful here and any suggested recommendations for future areas of research. Conclusion The conclusion in the abstract refers to 'silver linings' but I'm not
--	---

	sure what this relates to? Authors state that co-produced support for adolescents should be prioritised, but don't say how this might be achieved? References References to author's own work. Have these been published in peer reviewed journals? Appendices I was confused about the quotes in tables 1 and 2. Are these the quotes selected for inclusion in the paper? On what basis? Has a checklist for qualitative research been completed?
REVIEWER	Moss, Rachael Bradford Teaching Hospital NHS Foundation Trust
REVIEW RETURNED	02-Jan-2024
GENERAL COMMENTS	This is a really well written manuscript and was a very enjoyable read. It gives some really tangible methods of what could be addressed now we are in the post pandemic world too.

VERSION 1 – AUTHOR RESPONSE

Reviewer #1:

Stated Aims

The current interview and focus group study examines Covid-19's impact on 14-25-year-olds living with inequality and deprivation and co-produced support needs for this group.

This is an important focus but I am unclear how inequality and deprivation are defined in the context of the study? Can you say more about the criteria for selection? Or what you mean by co-produced support needs? Can you define co-production and explain in what sense the study was co-produced with young people?

Response: Thanks for your comment. We appreciate the reviewer's suggestion to contextualise the study and what we mean by 'inequality and deprivation', as well as the 'co-production process'. To address these comments, we have expanded the aims and objectives paragraph (lines 123-131, p. 5) and added more details on participants in the Methods section (lines 148-158, p. 6) to clarify the recruitment process of young people through Islington Council social workers working in Islington, one of the nation's 20% most deprived neighbourhoods with high unemployment rates (57.3% vs. rest of Britain, 39.8%).

It's unclear why and how participants felt ownership of the research project, as is claimed, when they were 'informed of the study aims' rather than developing these jointly with the research team.

Response: Thanks for your comment. Although the study aims were set from the get-go based on past studies of what young people from a similar background told the research team (Lenoir & Wong, 2023), the young people in the current study understood that the current set of interviews and focus groups were platforms for young people to express their needs and wants of how best to support them. They also expressed their appreciation for having a space to open up during interviews and informal discussions. Through this mutual understanding, young people could feel greater ownership of the project goals – they took part in the recruitment of participants (e.g. encouraging friends to join), contributed ideas for subsequent workshop content and delivery, and at later stages of the project, took part in designing ways to disseminate the study findings so that more young people could take part in the future.

Methods

Further detail would be helpful on how the study fits with the larger Empower-Islington partnership between University College London and Islington Council (<https://osf.io/82pgm/>) which aims to

identify, co-create, and assess the efficacy of life-skill workshops with Islington's 16-25-year-olds to maximise their life chances post-COVID.

Response: Thanks for your suggestion. We agree and have included more details on how the study fits within the wider Empower-Islington project on lines 134-139 (p. 5-6).

How and why were young people selected / recruited? What are the implications of the sample characteristics? e.g 12 women. Can you explain what is meant by a pre-registered study?

Response: Thanks for your question. The recruitment process was clarified and moved to the "Participants" section (lines 148-158, p. 6), to include the role of referrals through the Islington Council social workers and youth club workers (how) and reasons why they were recruited, as they were seen as young people who are hard-to-reach and NEET (not in education, employment or training) and could use additional support that the Council is not able to offer. We have also changed the wording of the participants: instead of using percentages (e.g. 60%) we've changed it to the number of participants (12 females) as per the reviewer's suggestion (line 160). We have also discussed the implications of our predominantly white and female sample (lines 167-170, p. 7).

To clarify what is meant by a 'pre-registered study', this is a pre-registration description of our study aims, design, and analytic plans to the Open Science Framework (<https://osf.io/82pgm/>) to ensure that steps are taken to reduce bias and fishing of results and data.

Design

It would be useful to include a clear rationale for the choice / order of methods used.

What were the strengths and limitations of having undergraduate and masters students conduct interviews and focus groups? Were they trained as peer researchers? What support did they receive before, during and after data collection? How were they (student researchers) recruited? Ethics of care?

Response: Thanks for the suggestions. To address this, we have added a new section ('Student researchers') detailing the recruitment process of and support from student peer researchers (see lines 172-177, p. 7), and the strengths and limitations of young student researchers conducting the interviews and focus groups (lines 178-184; lines 508-512, p. 19). That said, the lead researcher was also present in the focus groups.

Some bold claims are made about the ability to build rapport on the basis of age similarity. What about other experiences / inequalities, including gender, race/ethnicity, socioeconomic status, ability, sexual orientation? What is meant by 'sharing genuine experiences'?

Response: Thanks for your comment. We wanted to clarify that this is just an observation from the research team that being of similar age to the participants has made building rapport with participants a lot easier, but this is not the sole contributing factor for building rapport. To make this clearer in the paper, we have removed this claim on p. 3 ('strengths and limitations' and clarified this in Discussion (lines 476-479, p. 18) to reflect our observations. We instead write in the Abstract that a strength of the project is providing a space for young people to open up about their experiences of the pandemic with their peers (lines 68-69, p. 3).

Ethics

Did young people under 18 provide assent? How did you manage the ethics of working with young people experiencing inequalities?

Response: Thanks for flagging this. Yes, young people under 18 were only able to take part once parental consent was also received on the day of the interviews and focus groups – this change is now reflected in lines 156-158 (p. 6) under 'Participants'. We clarified that parental consent was received for under-18s and that parents were welcome to stay and wait in a separate space whilst interviews and focus groups were taking place to ensure inclusivity for those with additional needs, as was the case for some of our participants.

Public involvement

Aside from participating in interviews and focus groups, I am unclear how young people were involved, beyond helping with dissemination activities. If young people were not involved in the funding, design, RQs, analysis or interpretation of findings, please say why. If they were involved, please say how.

Response: Thanks for your comment. Apart from interviews and focus groups, young people were also actively involved in helping us recruit participants to join the interviews/focus groups, the designing of the content of the workshops themselves, dissemination of the workshops through their networks (including social media), feeding back on how the results should be shared, and worked closely with an animator for one of the project outputs. These changes are now reflected in lines 141-146 (p. 6) under 'public involvement'. We have also clarified this point in the Discussion. A limitation of our study is that we involved young people in interviews, focus groups, co-designing workshops and dissemination activities, but not in project research design, funding, and qualitative analysis (lines 503-506, p. 19).

Data analysis

What codes were generated? Could you include the coding framework in the appendix?

Response: Thanks for your question. Codes were generated on NVivo following transcription, then reorganised into themes following discussion with the research team (lines 197-203, p. 8). The coding framework is now added as Appendix C. Three 'parent' nodes were created: Covid-related, General, and Support Needs. Our manuscript focuses on Covid-related challenges and support needs; general developmental themes were not the focus hence not discussed in the manuscript. Eight main themes were generated around Covid-related and support needs and discussed in detail in Results.

I recommend Braun and Clark (2022) more recent book which encourages reflexive thematic analysis. I'd be interested in hearing more about the researchers' values, politics and positionality and how this influenced data collection and analysis and writing up.

Response: Thank you for your suggestion. We are grateful to the reviewer's suggestion of including a more critical reflection of how the researchers' positionality affects the current project. To address this point, we have included a paragraph in Methods on p. 7 (lines 178-184) with our reflections.

Themes

I was unclear how the themes were generated. For example, why support needs were not identified for all the themes, including grief / bereavement loss and loss of opportunities?

Response: Thanks for your comment. Participants discussed their need for support for some struggles (e.g. mental health, physical health struggles) but did not explicitly discuss support needs for other struggles, such as coping with grief/bereavement loss, distrust towards others, and loss of opportunities. As such, support needs were only identified for five out of eight themes. We have now clarified this in the first section of results (lines 207-210) and also under the concept map diagram (lines 213-214). Under themes 5, 7 and 8, we also included text clarifying that participants did not discuss support needs for them (lines 382, 436, 446-447).

The findings have important implications for policy and practice. How were young peoples' experiences differentiated by age, gender, ethnicity, experience of or contact with the CJS? Could these nuances be drawn out more explicitly in the findings and reflected in the discussion?

Response: Thanks for your comment. This is a great point. Whilst we do have information on young people's age, gender, and ethnicity – their experience of or contact with the CJS is not information that we have (as this resides with the case worker) and we agree this would be interesting to explore. As this was not a primary aim of our study and with the limited word count we have addressed this change by adding some more detail into the Discussion of findings in lines 480-485 (p. 18) – should the Editor be amenable to the additional word count this adds.

I would have welcomed further discussion about the interesting and important findings about trust / distrust of politicians and what this means for public health messaging and pandemic responses in the future.

Response: Thanks again for another helpful suggestion. We also agree with your point. To address this we have included additional text in Discussion on p. 17-18 (lines 464-467).

Findings

As a general rule, in small qualitative studies, it makes more sense to refer to numbers of participants rather than 90% or 60%. (as you do on P 13)

Response: Thanks for the helpful tip. We have now reflected this change throughout the manuscript. P14 refers to 'suboptimal welfare' without an explanation of what this means.

P17 lines 29-31. I don't understand the reference to county lines here. Can you explain?

Response: Thanks for your comment to clarify these statements. We have replaced 'suboptimal welfare' with 'lack of support and resources available' (line 357, p. 14) and 'county lines' with 'county lines – areas of the borough where criminal activities (e.g., drug crime) are known to take place.' (lines 496-497, p. 19) We hope this is now clearer.

Discussion

The discussion seems very short and doesn't appear to position the contribution the paper makes in existing literature or fully explore recommendations for practice. As noted, some reflections about the strengths and limitations of (peer research) methods would be helpful here and any suggested recommendations for future areas of research.

Response: Thanks for your comment. We have taken on board your suggestion and have now expanded the discussion to include more reflexive elements and implications for future research (e.g. focusing on specific groups, involving young people throughout the research process) on p. 19-20 (e.g. lines 483-485, 503-506, 517-518).

Conclusion

The conclusion in the abstract refers to 'silver linings' but I'm not sure what this relates to?

Response: Thanks for flagging this. We have now replaced 'silver linings' with 'positive reflections' (line 57, p. 3).

Authors state that co-produced support for adolescents should be prioritised, but don't say how this might be achieved?

Response: Thanks for your comment. We have now clarified this in the limitations and implications section of the Discussion (p. 20, e.g. lines 517-521, 530-532).

References

References to author's own work. Have these been published in peer reviewed journals?

Response: Yes, all of the references have been published in peer-reviewed journals.

Appendices

I was confused about the quotes in tables 1 and 2. Are these the quotes selected for inclusion in the paper? On what basis?

Response: The quotes are included in tables D1 and D2 (renamed) of Appendix D as they are representative of all available quotes on the same theme but could not all be listed in the manuscript. Has a checklist for qualitative research been completed?

Response: Thanks for spotting this. We have now completed the SRQR checklist for reporting of qualitative research and attached this to the revised manuscript.

Reviewer #2

This is a really well written manuscript and was a very enjoyable read. It gives some really tangible methods of what could be addressed now we are in the post pandemic world too.

Response: We are grateful to the reviewer for their encouraging comments, and hope that our manuscript will provide researchers and practitioners with tangible ideas of how to better engage and support young people four years on from the pandemic.